# Hypercoagulability in critically ill patients with COVID 19, an observational prospective study

**Laure Calvet[1], François Thouy[1], Olivier Mascle[1], Anne-Françoise Sapin[2], Kévin Grapin[1], Jean Mathias Liteaudon[1], Bertrand Evrard[3,4], Benjamin Bonnet[3,4], Mireille Adda[1], Bertrand Souweine[1,5], Claire Dupuis[1,6]***

1 Service de Médecine Intensive et Réanimation, CHU de Clermont-Ferrand, Clermont-Ferrand, France,
2 Laboratoire d'hématologie, CHU de Clermont-Ferrand, Clermont-Ferrand, France, 3 Service d'Immunologie, CHU Gabriel-Montpied, Clermont-Ferrand, France, 4 Laboratoire d'Immunologie, ECREIN, UMR1019 UNH, UFR Médecine de Clermont-Ferrand, Université Clermont Auvergne, Clermont-Ferrand, France, 5 Université Clermont Auvergne, CNRS, LMGE, Clermont-Ferrand, F-63000, France, 6 Université Clermont Auvergne, Unité de Nutrition Humaine, INRAe, CRNH Auvergne, Clermont-Ferrand, F-63000, France

* cdupuis1@chu-clermontferrand.fr

## Abstract

### Objective

COVID 19 is often associated with hypercoagulability and thromboembolic (TE) events. The aim of this study was to assess the characteristics of hypercoagulability and its relationship with new-onset TE events and the composite outcome of need for intubation and/or death in intensive care unit (ICU) patients admitted for COVID.

### Design

Prospective observational study.

### Setting

Monocentric, intensive care, University Hospital of Clermont Ferrand, France.

### Patients

Patients admitted to intensive care from January 2020 to May 2021 for COVID-19 pneumonia.

### Interventions

Standard hemostatic tests and rotational thromboelastometry (ROTEM) were performed on admission and on day 4. Hypercoagulability was defined by at least one of the following criteria: D-dimers > 3000 µg/dL, fibrinogen > 8 g/L, EXTEM CFT below the normal range, EXTEM A5, MCF, Li 60 above the normal range, and EXTEM G-score ((5000 x MCF) / (100-MCF)) $\geq$ 11 dyne/cm$^2$.

**Data Availability Statement:** All relevant data are within the paper and its Supporting Information files.

**Funding:** This study was supported by a grant from Michelin Corporate Foundation. The funders had no role in study design, data collection and analysis, decision to publish, or preparation of the manuscript.

**Competing interests:** No authors have competing interests.

## Measurements and main results

Of the 133 patients included, 17 (12.7%) developed new-onset TE events, and 59 (44.3%) required intubation and/or died in the ICU. ROTEM was performed in 133 patients on day 1 and in 67 on day 4. Hypercoagulability was present on day 1 in 115 (86.4%) patients. None of the hypercoagulability indices were associated with subsequent new-onset TE events on days 1 and 4 nor with the need for intubation and/or ICU death. Hyperfibrinogenemia > 8g/dL, higher D-dimers and higher EXTEM Li 60 on day 4 were predictive of need for intubation and/or of ICU death.

## Conclusions

Our study confirmed that most COVID-19 ICU patients have hypercoagulability on admission and almost all on day 4. Hyperfibrinogenemia or fibrinolysis shutdown on day 4 were associated with unfavorable outcome.

## Introduction

Patients with severe forms of COVID 19 have multifactorial hypercoagulability: increased D-dimer, fibrinogen, factor VIII levels, decreased protein C, protein S and antithrombin levels, platelet hyperaggregability, endothelial damage by Sars Cov 2 [1–3], and hypofibrinolysis. This hypercoagulability leads to thromboembolic events (TE) concerning macro and micro circulation [4–7]. These TE events, notably microcirculation thrombosis and pulmonary embolism, represent one of the leading causes of multiple organ failure in critically ill COVID-19 patients [4,6,8–14]. Thus, high dose preventive anticoagulation (HDPA) or even curative dose anticoagulation (CDA) were initially recommended in patients with severe forms of COVID-19 admitted to intensive care units [15–17].

In patients with COVID-19, several studies have reported that standard coagulation tests (SCT), such as platelet count, prothrombin time (PT), activated partial thromboplastin time (aPTT), and international normalized ratio (INR), may show normal results despite the presence of hypercoagulability [10]. Additionally, PT, aPTT, D-dimer and plasma fibrinogen assays do not explore fibrinolysis.

Rotational thromboelastometry (ROTEM) is a relocated laboratory assay providing a comprehensive analysis of dynamic and timely changes in the viscoelastic properties of clot formation, including clot initiation, propagation, strengthening, and dissolution. ROTEM tests assess both coagulation and fibrinolysis, are performed on whole blood and therefore take into account the contribution of blood cells. In ROTEM tests, clot formation can be triggered in various ways by the addition of different activators, defining different assays: EXTEM to assess the extrinsic pathway, INTEM the intrinsic pathway, FIBTEM the effect of fibrinogen on coagulation, and HEPTEM that of heparin on coagulation. In ROTEM assays, a probe is suspended within a sample of whole citrated blood, and changes in blood viscosity, which develop while the sample is rotated, are graphically transmitted. (S1 File).

ROTEM is recommended to guide the transfusion strategy in the management of major bleeding, since it can reduce blood product consumption [18–20]. It can also be performed to assess hypercoagulability [21–24], predict thrombosis in subgroups of patients at increased risk of TE events [25–28], and identify the effect of heparin on ROTEM results using heparinase (HEPTEM) [29]. Several studies performed on ROTEM in ICU patients admitted for

COVID reported hypercoagulability [30] and hypofibrinolysis [31–33] in patients with and without anticoagulant treatment that were present from the early phase of the disease [34], more severe in more acutely ill patients [35], and associated with unfavorable outcome [36].

Although several studies have attempted to establish a clear relationship between hypercoagulability and TE events and poor outcome in COVID-patients, such an association is still debated because of the small sample size of most studies, and variability in definitions of the procoagulant profile and of the clinical outcome assessment.

The aim of this study was to determine in critically ill patients admitted for COVID the prevalence of hypercoagulability on ICU admission and on day 4, compare the performance of standard coagulation and ROTEM tests for predicting TE events, and the unfavorable outcomes including need for intubation, ICU death, and the composite end point of need for intubation and/or ICU mortality.

## Methods

### Study population

This single center, prospective observational study was performed in the medical intensive care unit (ICU) of the university hospital of Clermont-Ferrand, France. All consecutive adult patients admitted between April 1, 2020 and March 31, 2021 for acute hypoxemic respiratory failure with a positive SARS COV 2 polymerase chain reaction (PCR)-based technique were included. CT pulmonary angiography was systematically performed on ICU admission. CT angiography (pulmonary/abdomino-pelvic/lower limbs/brain) was performed during ICU stay according to the evolution of clinical or laboratory parameters suggestive of TE events. TE events diagnosed before ICU admission or during ICU stay (new-onset TE event) were recorded. Patients were given antithrombotic treatment with LMWH or UFH using standard dose prophylactic anticoagulation (SPA), high dose prophylactic anticoagulation (HDPA), and curative dose anticoagulation (CDA) (S1 File) in accordance with the recommendations of the 'Groupe d'Etude sur l'Hémostase et la Thrombose'(GEHT) [15]. Demographic and clinical information, and laboratory and imaging data were extracted from the electronic charts. Vital status on ICU and hospital discharge were also recorded. Patients were managed without specific therapeutic intervention.

### Ethical considerations

All patients or their relatives received fair and relevant information. They gave written informed consent for the storage and research use of residual blood from samples collected as part of routine care. The study was approved by the ethics committee of the « Société de Réanimation de Langue Française » (SRLF) (CE SRLF n° 20–96) (CE French acronym for "commission d'éthique"). All the procedures were followed in accordance with the ethical standards of the responsible committee on human experimentation (institutional or regional) and with the Helsinki Declaration of 1975.

### Blood samples and laboratory features

Blood count and standard coagulation tests were performed daily during ICU stay including platelet count, activated partial thromboplastin time (aPTT) prothrombin time (PT), fibrinogen, and D-dimer (S1 File). On day 1, blood values of inflammatory parameters, including C-reactive protein (CRP), ferritin, interleukin-1 (IL1), IL-6, IL10, and monocyte HLA-DR (mHLA-DR) expression were measured as previously reported [37]. No patients received tocilizumab before day 1.

On day 1 and day 4 ± 1, two milliliters of residual blood from a citrate-containing tube were used to perform ROTEM tests in the ICU with a point-of-care testing device (ROTEM sigma, Werfen, Le pré saint Gervais, France).

The ROTEM tests, including extrinsic (EXTEM), intrinsic (INTEM), functional fibrinogen (FIBTEM), heparinase (HEPTEM) assays, were carried out in accordance with the manufacturer's instructions. The following ROTEM parameters were analyzed: clotting time (CT, s), clot formation time (CFT, s), α-angle and amplitude at 5 minutes (A5, mm), maximum clot firmness, (MCF, mm), and lysis index, the percentage of the remaining clot firmness in percentage of MCF at 60 minutes after CT (Li60, %).

## Definitions

TE events were defined by the onset of one of the following events: deep venous thrombosis (DVT), pulmonary embolism (PE), either catheter-related or not, ischemic stroke, systemic arterial thromboembolism, myocardial infarction, circuit clotting during renal replacement therapy (RRT), thrombosis in the extracorporeal membrane oxygenation (ECMO) circuit, diagnosed by ultrasound, computed tomography scan and ECG/Troponin, and clotting of the extracorporeal circuit.

Hypercoagulability was defined by at least one of the following indices: fibrinogen > 8 g/L, D-dimers > 3000 µg/dL, EXTEM CFT shorter than normal range, EXTEM A5 higher than normal range, EXTEM MCF higher than normal range, G score ≥11 dyne/cm$^2$, and EXTEM Li 60 > 96.5%. The EXTEM G-score was defined as follows: G-score = (5000xMCF) / (100-MCF). A major role of fibrinogen in a prothrombotic state was defined by a greater difference between observed and normal MCF values of FIBTEM than between those of EXTEM. Heparin was considered to affect ROTEM results when INTEM CT/ HEPTEM CT ratio was higher than 1.

Major bleeding events were recorded according to ISTH guidelines [38] (S1 File).

## Statistical analysis

Patient characteristics were expressed as n (%) for categorical variables and median (interquartile range [IQR]) for continuous variables. Comparisons were made with Student or Mann-Whitney tests and the Chi2 or Fisher exact tests for continuous and categorical data, respectively, depending on the distribution of the data. The correlations between all coagulation indices on admission were explored using coefficient correlations of Pearson or Spearman as appropriate.

Indices on admission and on day 4 were compared with paired t test or Wilcoxon test for paired data for continuous variables, and with McNemar's test for categorical data.

Factors associated with (1) the occurrence of TE events among patients free of TE events on admission, (2) intubation (excluding patients who died without intubation because of withholding of life support), (3) ICU death, and (4) the composite outcome including ICU death and/or intubation were investigated using Student or Wilcoxon tests, and the Chi2 or Fisher exact tests for continuous and categorical data, respectively, depending on the distribution of the data.

The ability of the coagulation indices to predict (1) TE events, (2) intubation, (3) ICU death, and (4) ICU death and/or intubation was assessed using the ROC curve. The best threshold value for the continuous variables was determined, and their corresponding area under the curve, sensibility and specificity were recorded.

A sensitivity analysis for missing data was performed to determine the amount and statistical nature of the missing data (informative and non-informative missing data: Missing At

Random, Missing Completely At Random, Not Missing At Random). Missing data were imputed using multiple imputation if needed.

A p value of < 0.05 was considered statistically significant. SAS® (Version 9.4; SAS Institute, Cary, NC, USA) and R (Version 3.4.0; R Core Team, Wien, Austria) softwares were used for the analysis.

## Results

### Baseline characteristics

During the study period, 133 patients were enrolled in the study (Table 1). In 23 patients, the medical background included TE events. In 10 patients, 7 PEs and 3 DVTs were diagnosed on ICU admission. In 17 (12.8%) patients, 21 new-onset TE events occurred during ICU stay (6 PEs, 2 DVTs, 1 catheter thrombosis, 1 acute lower limb ischemia, and 11 episodes of RRT filter clotting). Invasive mechanical ventilation was administered to 14 (10.6%) patients on day 1 and subsequently to 40 (31.2%) additional patients. ICU mortality rate was 24.8%. Ultimately, the composite outcome of intubation and/or ICU death was met by 59 (44.3%) patients.

### Laboratory parameters

The results of blood counts, standard coagulation and ROTEM tests on admission are shown in Table 2. On day 1, the presence of hypercoagulability was observed in 115 (86.4%) patients, with fibrinogen playing a major role in the increased clot firmness. No strong relationship was observed between routine blood coagulation parameters and ROTEM parameters (S1 Table and Fig 1). The analyses between inflammatory parameters and the indices of hypercoagulability are shown in S2 Table. No strong relationship was observed between standard coagulation or ROTEM tests and inflammatory parameters.

On day 4, ROTEM was performed in 67 patients. Of these, 66 (98.6%) had ≥ 1 indices of hypercoagulability, including 1 (1.4%) on SPA, 33 (48.6%) on HDPA and 28 (41.2%) on CDA. Of note, hypofibrinolysis was observed in 90.2% of patients. Fig 2 shows the results of ROTEM tests performed on admission in a patient with the graphical shape frequently observed showing both hypercoagulability and hypofibrinolysis.

The results of laboratory tests on day 1 and on day 4 in the 67 patients who underwent both tests are shown in S3 Table. On day 4, as compared to day 1, there was a higher percentage of patients with INTEM CT/HEPTEM CT ratio > 1 (70.2% vs 49.2%, p<0.01), and with ≥1 indices in favor of hypercoagulability (98.6% vs 88%, p = 0.02). Serum fibrinogen values decreased between day 1 and day 4, while all other indices of hypercoagulability had a value on day 4 in favor of a more pronounced state of hypercoagulability than on day 1, as shown by the increase in median values of the platelet count, G-score, EXTEM A5, EXTEM MCF, EXTEM Li60, and the decrease in median values of EXTEM CFT (S3 Table).

### Factors associated with TE events

None of the indices of hypercoagulability measured on day 1 or on day 4 were associated with new- onset TE events. Conversely, higher serum IL-6 values were (S4 Table). The ability of D-dimers, fibrinogen, platelet count, EXTEM CFT, A5, MCF, G-score, and Li 60 to predict the occurrence of new-onset TE events was assessed on days 1 and 4. All these parameters had a very low performance on day 1 (all AUC ROC curves < 0.7). On day 4, EXTEM Li 60 (AUC = 0.81 [0.74–0.88]), EXTEM MCF (AUC = 0.71 [0.48–0.93]), EXTEM G-score (AUC = 0.71 [0.48–0.93]) and platelet (AUC = 0.7 [0.58–0.83]) were the parameters associated with a greater ability to predict the occurrence of TE events (Fig 3, S5 Table).

**Table 1. Characteristics of patients on admission and during ICU stay.**

| Baseline Characteristics | All, n = 133 |
|---|---|
| Period of admission | Median(IQR), n(%) |
| March-August 2020 | 4 (3) |
| September- December 2020 | 75 (56.4) |
| January-June 2021 | 54 (40.6) |
| Age, median (IQR) | 71.2 [63.4; 76.4] |
| Gender (Male) | 95 (71.4) |
| BMI > 30 (kg/m$^2$) | 58 (43.6) |
| **Comorbidities** | |
| At least one | 73 (57.4) |
| Diabetes | 18 (13.6) |
| Cardiovascular disorder | 31 (23.4) |
| Lung disease | 8 (6) |
| Renal disease | 16 (12) |
| Immunosuppression§ | 19 (14.2) |
| Thromboembolic disease | 23 (17.2) |
| Time from onset of symptoms and ICU admission, days (missing = 12) | 8 [5; 10] |
| Immunomodulatory treatments before ICU admission§§ | 11 (8.3) |
| Anticoagulation before ICU admission | |
| None | 63 (47.4) |
| Prophylactic anticoagulation | 16 (12) |
| High-dose prophylactic anticoagulation* | 16 (12) |
| Curative anticoagulation | 38 (28.6) |
| Anti-platelet therapy before ICU admission | 36 (27) |
| **Severity on admission** | |
| SAPS II on admission | 35 [29; 44] |
| SOFA score | 4 [3; 6] |
| PaO2/FiO2 | 135 [116.6; 186] |
| IMV | 14 (10.6) |
| HFNC | 67 (50.4) |
| Thromboembolic event on admission | 10 (7.6) |
| **Other treatments on days 1 or 2** | |
| Vasopressors | 17 (12.8) |
| Renal replacement therapy | 3 (2.2) |
| Steroids on admission | 106 (79.6) |
| **Main events during ICU stay** | |
| IMV | 40 (31.2) |
| Vasopressors | 43 (33.6) |
| Renal replacement therapy | 18 (14) |
| Nosocomial infections | 33 (24.8) |
| Hemorrhagic events [Type I,II,III] | 15 (11.2)[2, 8, 5] |
| New onset thromboembolic event during ICU stay | 17 (12.8) |
| **Outcomes** | |
| Ventilatory-free days on day 28 | 28 [0; 28] |
| ICU length of stay | 7 [4; 14] |
| Hospital length of stay | 15 [8; 23] |
| ICU mortality | 33 (24.8) |

(*Continued*)

**Table 1.** (Continued)

| Baseline Characteristics | All, n = 133 |
|---|---|
| Hospital mortality | 46 (34.6) |
| Hospital death and/or invasive mechanical ventilation | 59 (44.4) |

§Immunosuppression concerned patients with long term or high dosage corticosteroid therapy, anticancer chemotherapy, AIDS, non-AIDS immunodepression, aplasia, bone or organ transplant recipient.

§§ Immunomodulatory treatments comprised IL1 anti receptor, IL6 anti receptor and steroids.

* High-dose prophylactic anticoagulation is defined by higher doses than the standard prophylactic regimen: Enoxaparin 6000 UI/ 24 h if BMI was below 30 kg/ $m^2$, enoxaparin 4000 UI/ 12 h if BMI was between 30 and 35 kg/ $m^2$, and enoxaparin 6000 UI/ 12 h if BMI was above 35 kg/$m^2$. Anti-Xa monitoring was performed if clearance was below 20 ml/min and if BMI was above 40 kg/$m^2$, in order to avoid overdose.

BMI: Body mass index; HFNC: High flow oxygen nasal cannula, ICU: Intensive care unit; IMV: Invasive mechanical ventilation; IQR: Interquartile range; SAPS: Simplified acute physiology score; SOFA: Sequential organ failure assessment; VFD: Ventilatory free days.

## Factors associated with intubation

The factors associated on day 1 with an increased risk of intubation were only higher serum IL-6 and Il-10 values and lower mHLA-DR. On day 4, high fibrinogen (p = 0.02) and high Li60 (p = 0.05) were associated with intubation during ICU stay. All the indices measured on days 1 or 4 had a very low ability (all AUC ROC curves < 0.7) to predict the need for intubation except for Li60 on day 4 (Fig 4, S6 and S7 Tables).

## Factors associated with ICU death

The factors associated on day 1 with an increased risk of ICU death was EXTEM Li60 (p = 0.05), and on day 4 higher were serum values of fibrinogen (p = 0.02), D-dimers (p = 0.04), and EXTEM Li 60 (p = 0.03). On day 1, higher serum IL-6 and Il-10 values and lower mHLA-DR values were associated with ICU death (S8 Table). All the indices measured on day 1 or 4 had a very low ability (all AUC ROC curves < 0.7) to predict the occurrence of ICU death (Fig 5, S9 Table).

## Factors associated with intubation and/or ICU death

The factors associated with an increased risk of intubation and/or ICU death on day 1 were INTEM CT/ HEPTEM CT ratio > 1 (p = 0.03), and on day 4 higher serum values of fibrinogen (p = 0.02), of D-dimers (p = 0.04), and of EXTEM Li 60 (p = 0.03). On day 1, higher serum IL-10 values and lower mHLA-DR values were associated with death or intubation (S10 Table). All the indices measured on day 1 or on day 4 had a very low ability (all AUC ROC curves < 0.7) to predict the occurrence of death and/or intubation (Fig 6, S10 and S11 Tables).

## Bleeding

Major bleeding complications were observed in 15 (11.30%) patients during ICU stay (Table 1). No bleeding led to death.

## Discussion

In this cohort, designed to characterize hypercoagulability in COVID-19 ICU patients using routine blood coagulation and ROTEM parameters, 21% of patients experienced a new TE event, 31% were intubated and 35% died in hospital. We confirmed hypercoagulability in

**Table 2. ROTEM, blood count and standard hemostasis on admission.**

| Laboratory features | Values, med(IQR)/N(%) |
|---|---|
| Number of patients | 133 |
| **Standard coagulation tests** | |
| PLATELETS, G/L | 254 [190; 315] |
| PROTHROMBIN TIME (TP), % | 84 [75; 92] |
| ACTIVATED PARTIAL THROMBOPLASTIN TIME (aPTT) | 1.2 [1; 1.4] |
| FIBRINOGEN, g/L | 7.2 [6.2; 8] |
| FIBRINOGEN > 8 g/L | 30 (26.6%) |
| D-DIMERS, µg/dL | 1188 [729; 2061.6] |
| D-DIMERS > 3000 µg/dL | 18(16.2%) |
| **ROTEM variables** | |
| EXTEM-CT, sec | 77 [70; 91] |
| EXTEM-CT, sec (>80 (Normal range)) | 62 (46.6) |
| EXTEM-CFT, sec | 48 [42; 56] |
| EXTEM-CFT, sec (<46 (Normal range)) | 61 (45.8) |
| EXTEM-A5, mm | 55 [50; 59] |
| EXTEM-A5, mm (> 52 (Normal range)) | 89 (67) |
| EXTEM MCF, mm | 73 [69; 75] |
| EXTEM MCF, mm (> 72 (Normal range)) | 78 (58.6) |
| EXTEM Li60, % (missing = 25) | 98 [95; 99] |
| EXTEM Li60, % (>96.5 (Normal range))(miss = 25) | 68 (63) |
| EXTEM G-score§ | 13.6 [11.2; 15] |
| EXTEM G-score§ > 11/ G score >14 | 102 (76.6)/58(43.6) |
| INTEM CT, sec | 186 [167; 206] |
| FIBTEM MCF, mm | *32 [28; 35]* |
| HEPTEM CT, sec | 184 [170; 201] |
| Hypercoagulation due to fibrinogenemia§§ | 131 (98.4) |
| INTEM CT / HEPTEM CT > 1 | 69 (51.8) |
| Presence of at least 1 index in favor of hypercoagulability§§§ | 115 (86.4) |
| Presence of at least 4 indices in favor of hypercoagulability | 49 (36.8) |

§ EXTEM G-score is defined as follows: G-score = (5000xMCF) / (100-MCF).

§§ Hypercoagulation due to fibrinogenemia was defined by a higher difference between observed and normal MCF values of FIBTEM than between observed and normal MCF values of EXTEM.

§§§ Hypercoagulability were defined by at least one of the following indices: Fibrinogen > 8 g/L, D-Dimers > 3000 µg/dL, EXTEM CFT shorter than normal range, EXTEM A5 higher than normal range, EXTEM MCF higher than normal range, G-score ≥11 dyne/cm$^2$, and EXTEM Li 60 > 96.5%.

CT: Clotting time; CFT: Clot formation time; A5: Clot amplitude 5 minutes after CT; MCF: Maximum clot firmness; Li60: Lysis index at 60 minutes.

COVID with fibrinogen playing a predominant role. ROTEM indices were not strongly correlated with results of standard blood tests. EXTEM Li 60, EXTEM MCF, EXTEM G-score and platelet were the most reliable indices to predict the occurrence of TE events, whereas hyperfibrinogenemia, higher D-dimer and higher EXTEM Li 60 levels on day 4 were associated with an increased risk of death and intubation. Finally, anticoagulation on admission as evidenced by an INTEM-CT/ HEPTEM-CT >1 was associated with an increased risk of death or intubation.

We observed a lower rate of TE events in earlier studies, which occurred in 30 to 35% of COVID-19 ICU patients and were mainly cases of pulmonary embolism [39–41]. These

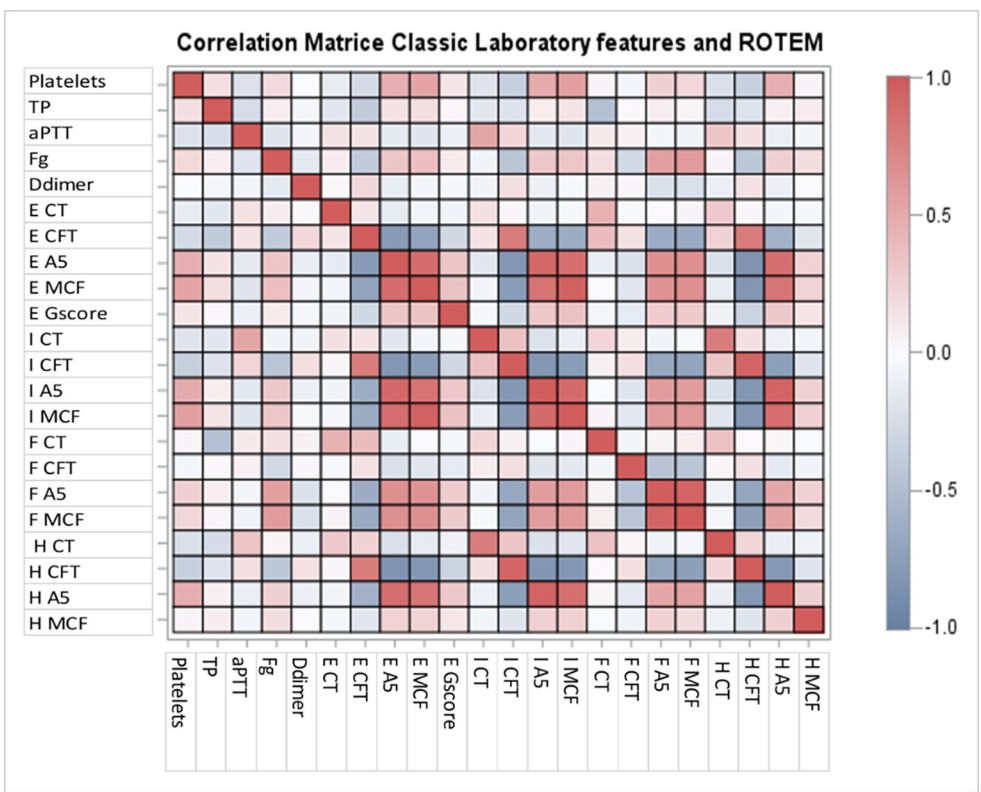

**Fig 1. Matrix correlation of the indices from routine coagulation and ROTEM on admission.** TP: Prothrombin Time; aPTT: Activated Partial Thromboplastin Time; Fg: Fibrinogen; E: EXTEM; I: INTEM; F: FIBTEM; H: HEPTEM; CT: Clotting time; CFT: Clot formation time; A5: Clot amplitude 5 minutes after CT; MCF: Maximum clot firmness.

figures were confirmed in a recent meta-analysis of 15 studies that recorded a 27.9% rate of venous TE events among COVID-19 ICU patients [42]. However, a recent large study also found lower rates of TE events, of 6.4% and 10.4%, in patients with preventive and curative anticoagulation, respectively [43]. This lower rate could be explained by the study period: most patients were enrolled after July 2020 and were prescribed high-dose prophylactic or curative anticoagulation, as then recommended [5,15,17,44–46], and also steroids, which could have minimized inflammation and associated hypercoagulation [47]. In addition, since follow-up CT scans and Doppler ultrasound were only performed in our ICU when suspected from the evolution of clinical or laboratory parameters, we cannot exclude that we underdiagnosed the occurrence of a new TE event. In our study, like most authors, we found no strong correlations between standard blood tests and ROTEM [48–50].

ROTEM showed biological hypercoagulability in our COVID-19 ICU patients on days 1 and 4, which combined increased clot propagation and firmness, and hypofibrinolysis. Similar results have been previously reported in the ICU [13,35,51–55]. For instance, Saseedharan et al. [56] and Yuriditsky et al.[51] retrospectively observed hypercoagulation based on the thromboelastography profile of 62.5% and 50% among their cohorts of 32 and 64 COVID-19 ICU patients, respectively. Almskog et al. found a hypercoagulation thromboelastometry profile with EXTEM-MCF and FIBTEM-MCF significantly higher in 60 hospitalized COVID-19 patients (including 20 ICU patients) than in healthy volunteers. A higher state of hypercoagulability on ROTEM profiles was also reported in COVID patients admitted to the ICU than in less severely ill patients admitted to medical wards [35,57].

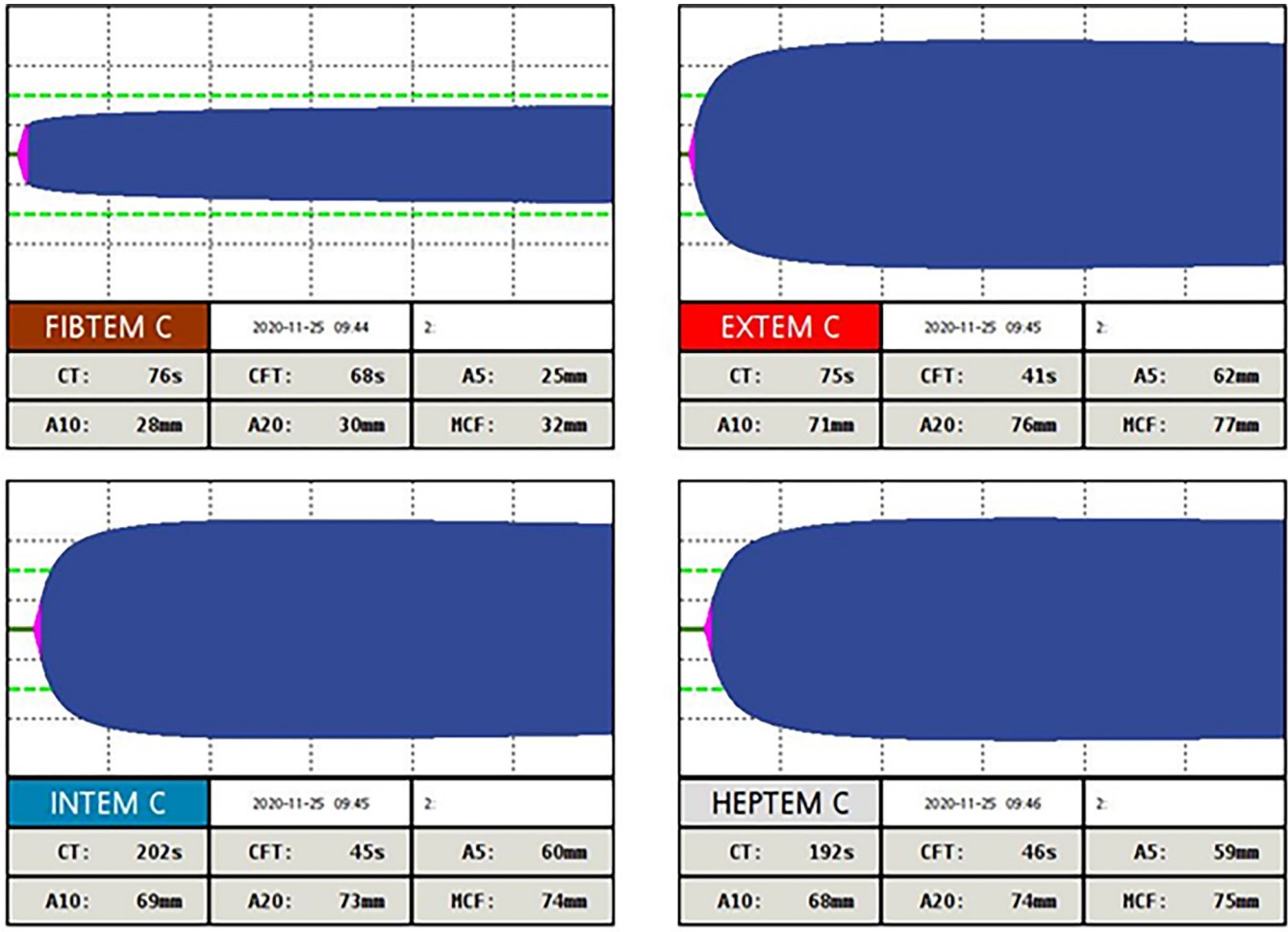

**Fig 2. Results of ROTEM tests with the graphical shape performed on admission showing both hypercoagulability and hypofibrinolysis.** In this patient, hypercoagulability was characterized by an EXTEM CFT result below normal range values (< 46 s), and EXTEM A5 and EXTEM MCF results above normal range values (> 52 mm and > 72 mm, respectively); hypofibrinolysis was characterized by EXTEM Li 60 > 96.5%. CT: Clotting time; CFT: Clot formation time; A5 –A10 –A20: Clot amplitude 5–10–20 minutes after CT; MCF: Maximum clot firmness.

In our study we observed that hypercoagulation was mostly due to both hyperfibrinogenemia, and hypofibrinolysis. Although not systematically reported by authors [13,58], fibrinolysis shutdown has been documented as playing a predominant role in COVID-19 hypercoagulability [59–63], driven either by COVID or by the severity of the patient's condition [64].

In our study, among the hypercoagulability indices, hypofibrinolysis on day 4 as defined by elevated Li60 values had the best accuracy to predict new-onset TE events. The association between fibrinolysis shutdown and risk of TE events in COVID-19 ICU patients has been previously reported. In a prospective study of 44 ICU patients, Wright et al. described an association between thromboelastography fibrinolysis shutdown defined by a zero value of the Li30 parameter and venous TE events (AUC 0.742). In addition, they proposed a combined score of Li30 values and D-dimer levels to estimate the risk of venous TE events during ICU stay [59]. Similarly, in a prospective study of 40 COVID-19 ICU patients, Kruse et al. reported an exacerbated venous TE risk associated with low EXTEM and INTEM maximum lysis (ML) values, with AUCs for both tests of 0.8. The authors also reported that the difference between EXTEM

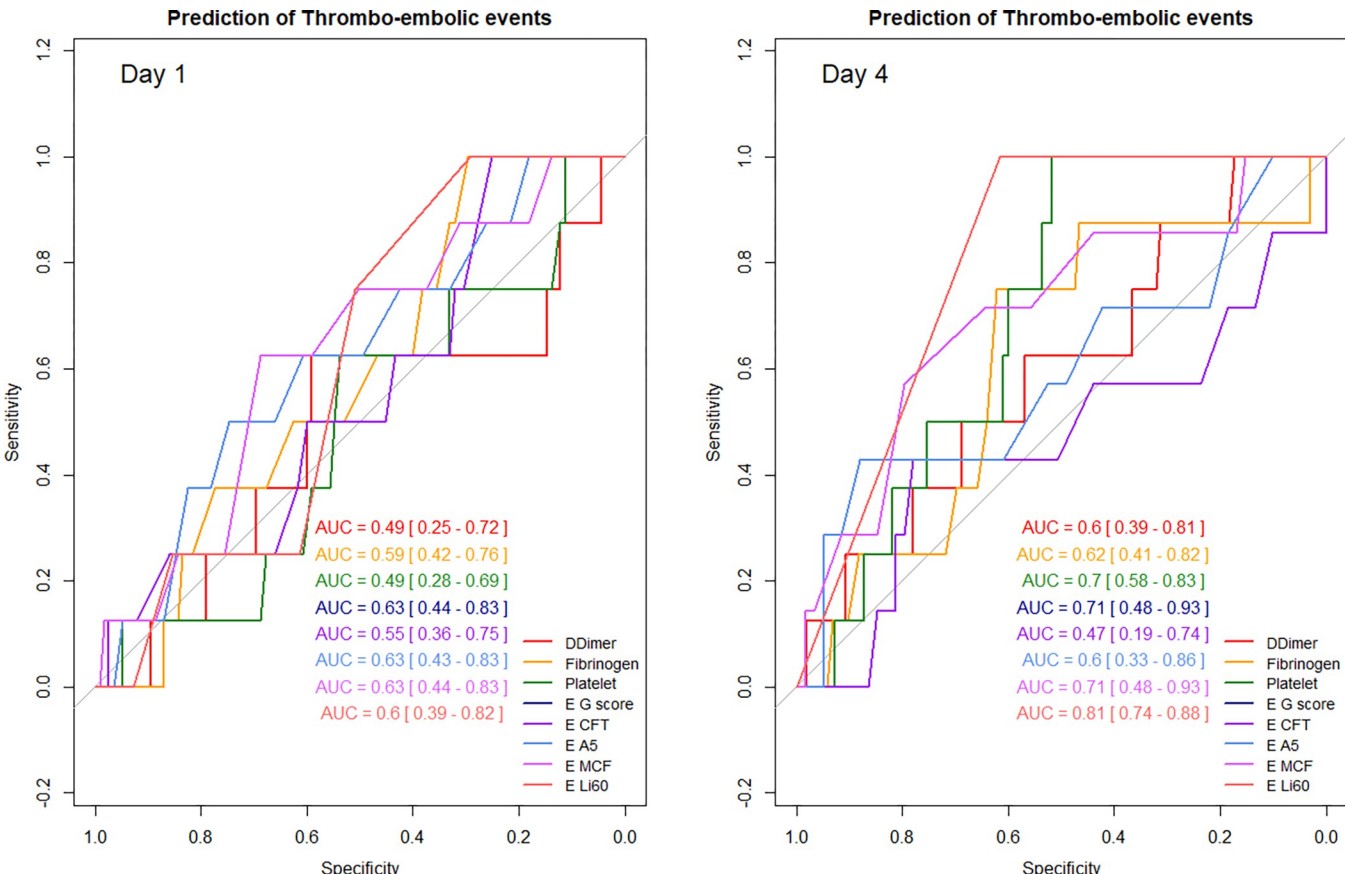

**Fig 3. Prediction of the occurrence of thrombo-embolic events during ICU stay by coagulation indices on day 1 and day 4.** AUC: Area Under the Curve; E: EXTEM; CFT: Clot formation time; A5: Clot amplitude 5 minutes after CT; MCF: Maximum clot firmness; Li60: Lysis index at 60 minutes.

ML and maximum D-dimer levels predicted TE risk with an AUC of 0.9 [63]. Taken together these findings support a predominant role of reduced fibrinolysis in the development of venous TE events in severe COVID-19. In our study, serum fibrinogen and D-dimer levels, in contrast to Li60 values, failed to predict TE events, underlining the better sensitivity of ROTEM indices over standard blood tests in this setting.

On day 4, hyperfibrinogenemia, and elevated D-dimer levels and Li60 values, which can all reflect greater fibrinolysis resistance, were associated with an increased risk of intubation and/ or ICU death. In COVID ICU patients, the association between unfavorable outcome and elevated fibrinogen [65] and D-dimer levels [66–68] has been previously reported. Similarly, Roh et al. suggested that the significant increase in fibrinogen plasma concentrations and FIBTEM MCF indicates the severity of COVID-19 and can be used for risk stratification for thrombosis, respiratory failure, and mortality [66]. The relationship between Li60 values and mortality remains debatable [32]: it has been reported in septic patients, but never to our knowledge in COVID-19 ICU patients [36].

In our study, we observed prolonged clot initiation. Whether it can be related to heparin exposure requires a careful analysis since the impact of heparin on INTEM-CT depends on both the type of heparin and its dosage. While ROTEM tests are able to detect the presence of UFH on the basis of an increase in the difference, or in the ratio, between INTEM-CT and HEPTEM-CT, and whereas in patients administered with UFH a prolonged INTEM-CT is

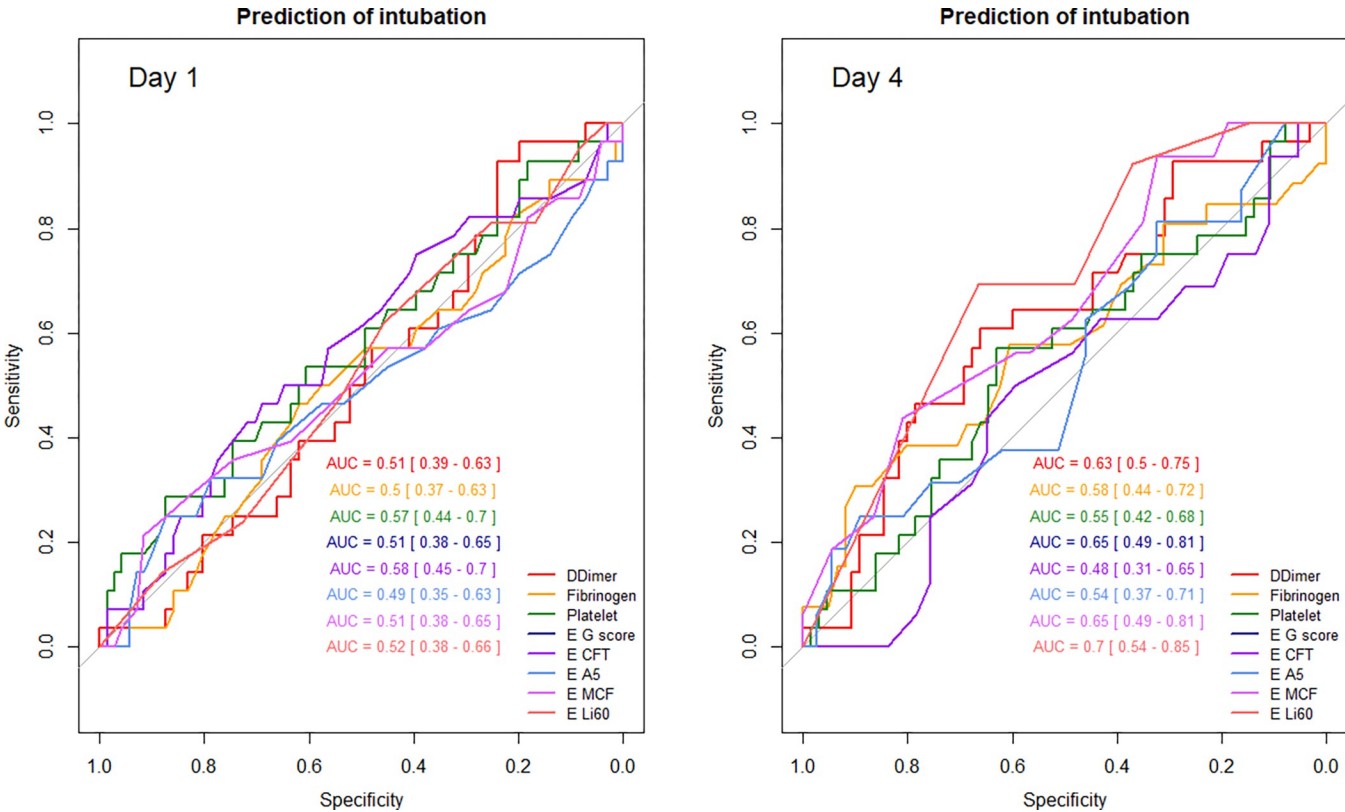

**Fig 4. Prediction of intubation during ICU stay by coagulation indices on day 1 and day 4.** AUC: Area Under the Curve; E: EXTEM; CFT: Clot formation time; A5: Clot amplitude 5 minutes after clotting time; MCF: Maximum clot firmness; Li60: Lysis index at 60 minutes.

reported for heparinemia > 0.1 U/l, both traditional coagulative parameters and ROTEM profiles are poorly influenced by LMWH, and an increased INTEM-CT in patients receiving LMWH is only observed for heparinemia >0.4UI/l [69,70]. We observed similar results in our study. INTEM/HEPTEM CT > 1 was detected in patients receiving UFH for HDPA or CDA, and in 90% of patients under LMWH for CDA, but only in half of the patients under LMWH for HDPA (S12 Table).

Despite HDPA and even CDA, hypercoagulability was more pronounced on day 4 than on day 1. These results are in close agreement with those of previous studies [32,71]. Our data support the results of REMAP-CAP demonstrating the absence of benefit of CDA in COVID ICU patients [43]. One possible explanation is that CDA was given too late to reverse the consequences of the disease process in severe COVID-19 patients. In line with these results, we also observed that an INTEM-CT longer than an HEPTEM-CT, which was mostly observed in patients administered with HDPA or CDA UFH, or with CDA LMWH, was associated with an increased risk of intubation and/or ICU death. In our cohort, CDA was administered to patients with documented TE events and in those with elevated D-dimer and/or fibrinogen levels. We cannot therefore exclude that these criteria identify the most severely ill patients. Furthermore, in our cohort, most of the time, the patients under UFH are the most severely ill patients, i.e., those receiving mechanical ventilation or renal replacement therapy. This could also contribute to the relationship between the ratio INTEM/HEPTEM CT > 1 and worse outcomes in our cohort.

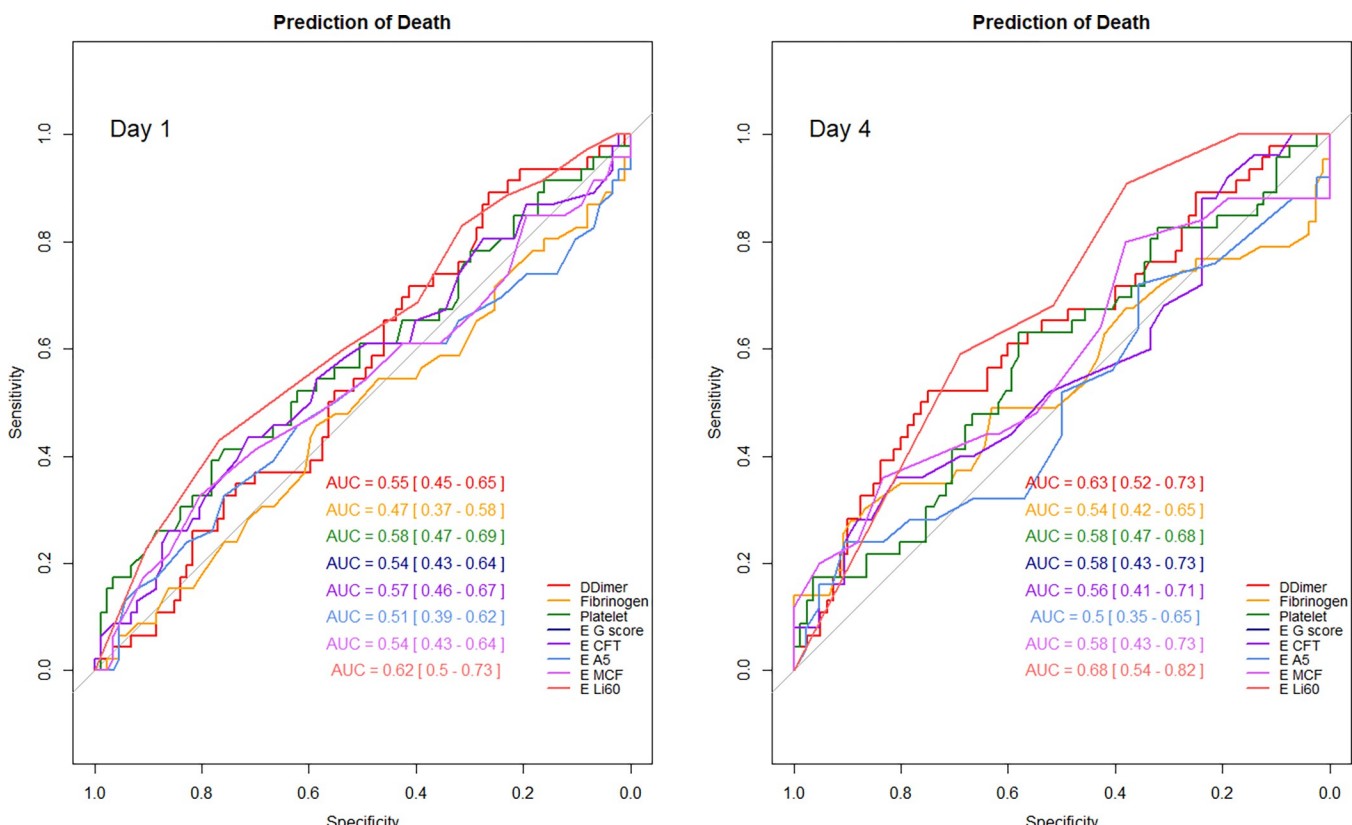

**Fig 5. Prediction of ICU death by coagulation indices on day 1 and day 4.** AUC: Area Under the Curve; E: EXTEM; CFT: Clot formation time; A5: Clot amplitude 5 minutes after clotting time; MCF: Maximum clot firmness; Li60: Lysis index at 60 minutes.

However, even though major bleeding events were few and unrelated to CDA, it is possible that in some patients CDA had been harmful by exacerbating alveolar hemorrhage.

The main strength of this study is to be one of the largest cohorts of critically ill COVID patients monitored by ROTEM on admission and on day 4. However, it also has several limitations. First, it was a monocentric study involving a cohort of patients admitted during the second and third waves in France, and therefore most of the patients were on steroids and receiving protocolized anticoagulation, which limits comparisons with the first published studies. Second, ROTEM was performed on day 4 in only 60% of our patients, mostly because of death or living alive patients. Third, we chose day 4 as the onset of inflammation whereas other authors proposed day 7 [37]. Finally, several TE events might have been missed because of the retrospective recording of their occurrence.

## Conclusions

Our study confirmed that a majority of critically ill COVID-19 patients had hypercoagulability as a result of hyperfibrinogenemia mostly secondary to hypofibrinolysis. Routine blood coagulation and ROTEM parameters were poorly correlated. On day 4, the accuracy of Li60 to predict new onset TE events was good, and hyperfibrinogenemia and Li60 values were good indices to identify the patients the most at risk of intubation and/or ICU death. Our findings question the systematic curative anticoagulation strategy proposed in COVID-19 patients with elevated fibrinogen and/or D-dimer levels.

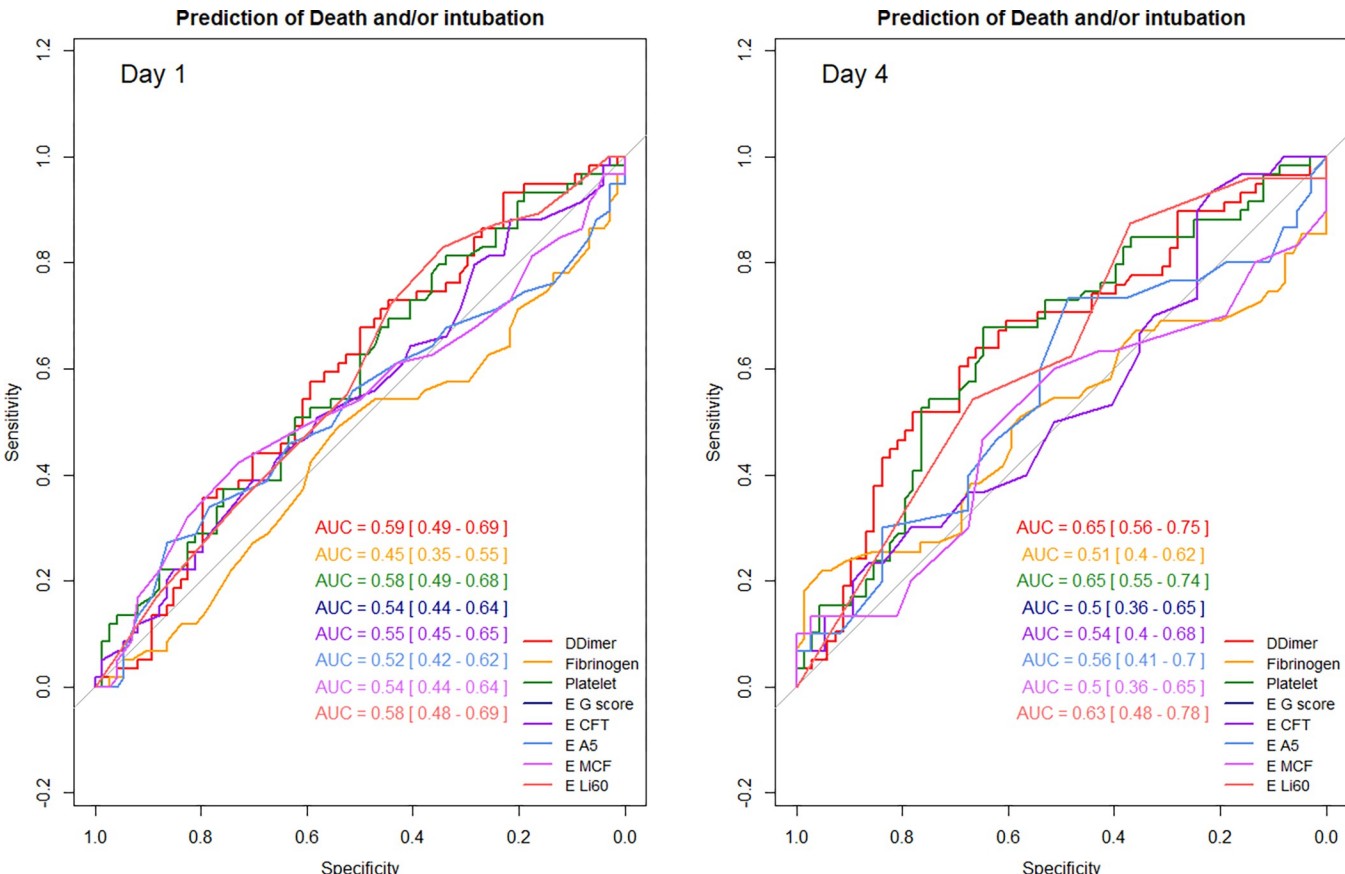

**Fig 6. Prediction of intubation and/or ICU death by coagulation indices on day 1 and day 4.** AUC: Area Under the Curve; E: EXTEM; CFT: Clot formation time; A5: Clot amplitude 5 minutes after clotting time; MCF: Maximum clot firmness; Li60: Lysis index at 60 minutes.

## Supporting information

**S1 Table. Coefficient correlation matrix between indices from ROTEM and others from usual laboratory features.** TP: Prothrombin Time; aPTT: Activated Partial Thromboplastin Time; Fg: Fibrinogen; E: EXTEM; I: INTEM; F: FIBTEM; H: HEPTEM; CT: Clotting time; CFT: Clot formation time; A5: Clot amplitude at 5 minutes; MCF: Maximum clot firmness. (DOCX)

**S2 Table. Coefficient correlation matrix between indices of hypercoagulability and inflammatory parameters.** CRP: C reactive protein; Fg: Fibrinogen; E: EXTEM; CFT: Clot formation time; A5: Clot amplitude at 5 minutes; MCF: Maximum clot firmness; Li60: Lysis index at 60 minutes. (DOCX)

**S3 Table. Comparison between day 1 and day 4 for standard hemostasis and ROTEM.** § EXTEM G-score is defined as follows: G-score = (5000xMCF) / (100-MCF). §§ Hypercoagulation due to fibrinogenemia was defined by a higher difference between observed and normal MCF values of FIBTEM than between observed and normal MCF values of EXTEM. §§§ Hypercoagulability were defined by at least one of the following indices: Fibrinogen > 8 g/L, D-Dimers > 3000 µg/dL, EXTEM CFT shorter than normal range, EXTEM A5 higher than normal range, EXTEM MCF higher than normal range, G score ≥11 dyne/cm2, and EXTEM

Li 60 > 96.5%. CFT: Clot formation time; A5: Clot amplitude at 5 minutes; MCF: Maximum clot firmness; Li60: Lysis index at 60 minutes.
(DOCX)

**S4 Table. Factors associated with the occurrence of thrombo-embolic events during intensive care stay.** TE: Thrombo-embolic event; CFT: Clot formation time; A5: Clot amplitude at 5 minutes; MCF: Maximum clot firmness; Li60: Lysis index at 6 minutes; CRP: C reactive protein.
(DOCX)

**S5 Table. Prediction of occurrence of thrombo-embolic events during ICU stay by coagulation indices on days 1 and 4.** AUC: Area under the curve; CFT: Clot formation time; A5: Clot amplitude at 5 minutes; MCF: Maximum clot firmness; Li60: Lysis index at 60 minutes.
(DOCX)

**S6 Table. Factors associated with intubation during ICU stay among patients non intubated on admission.** CFT: Clot formation time; A5: Clot amplitude at 5 minutes; MCF: Maximum clot firmness; Li60: Lysis index at 60 minutes.
(DOCX)

**S7 Table. Prediction of intubation during ICU stay by coagulation indices on days 1 and 4.** AUC: Area under the curve; CFT clot formation time; A5: Clot amplitude at 5 minutes; MCF: Maximum clot formation time; Li60: Lysis index at 60 minutes.
(DOCX)

**S8 Table. Comparison of patients who experienced ICU death and those who did not.** CFT: Clot formation time; A5: Clot amplitude at 5 minutes; MCF: Maximum clot firmness; Li60: Lysis index at 60 minutes.
(DOCX)

**S9 Table. Prediction of occurrence of death by coagulation indices on days 1 and 4.** AUC: Area under the curve; CFT clot formation time; A5: Clot amplitude at 5 minutes; MCF: Maximum clot formation time; Li60: Lysis index at 60 minutes.
(DOCX)

**S10 Table. Comparison of patients who experienced hospital death and/or invasive mechanical ventilation and those who did not.** CFT: Clot formation time; A5: Clot amplitude at 5 minutes; MCF: Maximum clot firmness; Li60: Lysis index at 60 minutes.
(DOCX)

**S11 Table. Prediction of occurrence of death and/or intubation by coagulation indices on days 1 and 4.** AUC: Area under the curve; CFT clot formation time; A5: Clot amplitude at 5 minutes; MCF: Maximum clot formation time; Li60: Lysis index at 60 minutes.
(DOCX)

**S12 Table. Anticoagulation and prolonged CT on day 4.** HDPA: High dose preventive anticoagulation; CDA: Curative dose anticoagulation.
(DOCX)

**S1 File. Supplemental file–definitions.**
(DOCX)

**S1 Data.**
(XLSX)

## Acknowledgments

We thank JeffreyWatts for advice on the English manuscript.

## Author Contributions

**Conceptualization:** Laure Calvet, François Thouy, Anne-Françoise Sapin, Mireille Adda, Bertrand Souweine, Claire Dupuis.

**Data curation:** François Thouy, Anne-Françoise Sapin, Kévin Grapin, Jean Mathias Liteaudon, Bertrand Evrard, Benjamin Bonnet, Mireille Adda, Claire Dupuis.

**Formal analysis:** Anne-Françoise Sapin, Benjamin Bonnet, Mireille Adda, Claire Dupuis.

**Funding acquisition:** Laure Calvet, Mireille Adda, Bertrand Souweine.

**Investigation:** Laure Calvet, Bertrand Souweine, Claire Dupuis.

**Methodology:** Laure Calvet, Anne-Françoise Sapin, Bertrand Souweine, Claire Dupuis.

**Project administration:** Bertrand Souweine.

**Resources:** François Thouy, Mireille Adda, Bertrand Souweine.

**Software:** Claire Dupuis.

**Supervision:** Bertrand Souweine, Claire Dupuis.

**Validation:** Bertrand Evrard, Bertrand Souweine, Claire Dupuis.

**Visualization:** Laure Calvet, Olivier Mascle, Anne-Françoise Sapin, Claire Dupuis.

**Writing – original draft:** Laure Calvet, Olivier Mascle, Bertrand Souweine, Claire Dupuis.

**Writing – review & editing:** François Thouy, Olivier Mascle, Anne-Françoise Sapin, Kévin Grapin, Jean Mathias Liteaudon, Bertrand Evrard, Benjamin Bonnet, Mireille Adda.

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
