## [Decision Letter · Decision Letter 0]

22 Sep 2022

PONE-D-22-24018Hypercoagulability in critically ill patients with COVID 19, an observational prospective study.PLOS ONE

Dear Dr. Dupuis,

Thank you for submitting your manuscript to PLOS ONE. After careful consideration, we feel that it has merit but does not fully meet PLOS ONE’s publication criteria as it currently stands. Therefore, we invite you to submit a revised version of the manuscript that addresses the points raised during the review process.

ACADEMIC EDITOR:Dear authors,

the paper looks interesting and may add a further step in understanding the negative impact of the hypercoagulable state found in critically ill patients with COVID-19. I think it is a good paper, well written, but in need of minor revisions as requested by the Reviewers, especially the review of the outcome as per Reviewer #1. As far as the thromboelastogram is concerned, in addition to a small and brief explanation (no more than 1-2 sentences), it would be useful to include in the main paper a meaningful picture of the ROTEM, showing at a glance the picture that is most frequently seen in these patients.

We look forward to receiving your revised manuscript.

Kind regards,

Samuele Ceruti

Academic Editor

PLOS ONE

"We thank the Michelin Corporate Foundation for their grant"

"This study was supported by a grant from Michelin Corporate Foundation.

Reviewers' comments:

Reviewer's Responses to Questions

**Comments to the Author**

1. Is the manuscript technically sound, and do the data support the conclusions?

Reviewer #1: Yes

Reviewer #2: Yes

2. Has the statistical analysis been performed appropriately and rigorously? 

Reviewer #1: Yes

Reviewer #2: Yes

3. Have the authors made all data underlying the findings in their manuscript fully available?

Reviewer #1: Yes

Reviewer #2: Yes

4. Is the manuscript presented in an intelligible fashion and written in standard English?

Reviewer #1: Yes

Reviewer #2: Yes

5. Review Comments to the Author

Reviewer #1: Several studies published in the Literature so far reported a hypercoagulable profile in patients with acute COVID19-related pneumonia. However, there are still few studies that have tried to highlight a possible correlation between alteration of coagulation parameters and bad outcomes. The results of the study by Calvet et al. help bridge this gap. The study is clear and well written. The results are solid and the conclusions reported by the authors are in agreement with the data presented in the study. The only aspect on which I disagree is that of having associated the two end points "need for intubation" and "death in intensive care unit" in a single outcome. I believe that it is necessary to separately identify the coagulation profile that characterizes each of these two groups of patients and, consequently, to separately analyze the possible alterations of the coagulation profile for each of the two groups.

Minor revisions: check the text for typos (i.e. abstract line 2 "thromboembolic (TE) events" change in "thromboembolic events (TE)".

Reviewer #2: The manuscript of Calvet et al. evaluates the coagulation profile according to both traditional coagulation tests and thromboelastometry parameters in a group of 133 patients admitted for acute COVID-19 pneumonia. Interestingly enough the authors tried to find a correlation between the alterations of coagulation parameters and a composite endpoint "intubation and / or death in intensive care unit". The study results demonstrate a possible association between hyperfibrinogenemia or fibrinolysis shutdown on day 4 and the composite outcome "intubation and / or death in intensive care unit".

The results presented in the paper support the study's conclusions. Statistical analysis is appropriate and rigorous.

My comments:

Abstract: "despite high use of therapeutic anticoagulation" I do not agree with this concept since both traditional coagulative parameters and ROTEM profiles are poorly influenced by low molecular weight heparin

Introduction: the authors should explain in more detail the relationship between COVID-19 related pneumonia, hypercoagulability and hypofibrinolysis

Methods: the authors should better explain the principle of thromboelastometry and of the assays used.

Discussion: line 3 pag. 13 "Finally, we observed prolonged clot initiation on day 1 presumably owing to heparin exposure" see my comment above in the "abstract section".

6. PLOS authors have the option to publish the peer review history of their article (what does this mean?). If published, this will include your full peer review and any attached files.

Reviewer #1: No

Reviewer #2: No

---

## [Author Response · Author response to Decision Letter 0]

14 Oct 2022

Response to reviewers: 

ACADEMIC EDITOR:

Dear authors,

the paper looks interesting and may add a further step in understanding the negative impact of the hypercoagulable state found in critically ill patients with COVID-19. I think it is a good paper, well written, but in need of minor revisions as requested by the Reviewers, especially the review of the outcome as per Reviewer #1. 

As requested, we analyzed separately the alteration of coagulation profile according to (1) thromboembolic events, (2) need for intubation, (3) ICU death, and (4) the composite outcome of intubation and/or death. 

Since in several patients withholding of life support resulted in ICU death without intubation, in the analysis of the relationship between coagulation parameters and the need for intubation, the patients who died and were not intubated were excluded. 

This is indicated at the end of the introduction section: 

“ROTEM tests for predicting TE events, and the unfavorable outcomes including need for intubation, ICU death, and the composite end point of need for intubation and/or ICU mortality” 

In the Statistical analysis

“Factors associated with (1) the occurrence of TE events among patients free of TE events on admission, (2) intubation (excluding patients who died without intubation because of withholding of life support), (3) ICU death, and (4) the composite outcome including ICU death and/or intubation were investigated using Student or Wilcoxon tests, and the Chi2 or Fisher exact tests for continuous and categorical data, respectively, depending on the distribution of the data.

The ability of the coagulation indices to predict (1) TE events, (2) intubation, (3) ICU death, and (4) ICU death and/or intubation was assessed using the ROC curve.”

In the results section

“Factors associated with intubation 

The factors associated on day 1 with an increased risk of intubation were… 

All the indices measured on days 1 or 4 had a very low ability (all AUC ROC curves < 0.7) to predict the need for intubation (Fig 4, S7 Table).

Factors associated with ICU death 

The factors associated on day 1 with an increased risk of ICU death was EXTEM Li60 (p=0.05), and on day 4 higher were serum values of fibrinogen (p=0.02), D-dimers (p = 0.04), and EXTEM Li 60 (p = 0.03). On day 1, higher serum IL-6 and Il-10 values and lower mHLA-DR values were associated with ICU death (S8 Table). All the indices measured on day 1 or 4 had a very low ability (all AUC ROC curves < 0.7) to predict the occurrence of ICU death (Fig 5, S9 Table).

As far as the thromboelastogram is concerned, in addition to a small and brief explanation (no more than 1-2 sentences), it would be useful to include in the main paper a meaningful picture of the ROTEM, showing at a glance the picture that is most frequently seen in these patients.

Thank you for this remark. 

First, we added this sentence in the introduction 

“Rotational thromboelastometry (ROTEM) is a relocated laboratory assay providing a comprehensive analysis of dynamic and timely changes in the viscoelastic properties of clot formation, including clot initiation, propagation, strengthening, and dissolution. ROTEM tests assess both coagulation and fibrinolysis, are performed on whole blood and therefore take into account the contribution of blood cells. In ROTEM tests, clot formation can be triggered in various ways by the addition of different activators, defining different assays: EXTEM to assess the extrinsic pathway, INTEM the intrinsic pathway, FIBTEM the effect of fibrinogen on coagulation, and HEPTEM that of heparin on coagulation. In ROTEM assays, a probe is suspended within a sample of whole citrated blood, and changes in blood viscosity, which develop while the sample is rotated, are graphically transmitted. (S File). (S File).” 

As requested, a meaningful picture of the ROTEM, showing at a glance the picture that is frequently observed in these patients has been included in the paper (Figure 2)

In the results section: 

Figure 2 shows the results of ROTEM tests performed on admission in a patient with the graphical shape frequently observed showing both hypercoagulability and hypofibrinolysis.

Fig 2: Results of ROTEM tests with the graphical shape performed on admission showing both hypercoagulability and hypofibrinolysis.

 CT: clotting time; CFT: clot formation time; A5 – A10 – A20: clot amplitude 5 – 10 – 20 minutes after CT; MCF: maximum clot firmness

In this patient, hypercoagulability was characterized by an EXTEM CFT result below normal range values (< 46s), and EXTEM A5 and EXTEM MCF results above normal range values (> 52 mm and > 72 mm, respectively); hypofibrinolysis was characterized by EXTEM Li 60 > 96.5%.

Ok done

"This study was supported by a grant from Michelin Corporate Foundation.

Ok, thank you for this suggestion. We leave it to you to place the above sentence appropriately in the manuscript. 

We have left our sentence in the Funding paragraph of the manuscript and added this sentence in the cover letter. 

We have removed from the Acknowledgements section the sentence about the Michelin Corporation

OK, all the data are now available publicly. 

Reviewers' comments:

Reviewer's Responses to Questions

Comments to the Author

5. Review Comments to the Author

Reviewer #1: Several studies published in the Literature so far reported a hypercoagulable profile in patients with acute COVID19-related pneumonia. However, there are still few studies that have tried to highlight a possible correlation between alteration of coagulation parameters and bad outcomes. The results of the study by Calvet et al. help bridge this gap. The study is clear and well written. The results are solid and the conclusions reported by the authors are in agreement with the data presented in the study. The only aspect on which I disagree is that of having associated the two end points "need for intubation" and "death in intensive care unit" in a single outcome. I believe that it is necessary to separately identify the coagulation profile that characterizes each of these two groups of patients and, consequently, to separately analyze the possible alterations of the coagulation profile for each of the two groups.

The request has been taken into account (see reply above to the Academic Editor)

Minor revisions: check the text for typos (i.e. abstract line 2 "thromboembolic (TE) events" change in "thromboembolic events (TE)". 

All typos in the manuscript were corrected for thromboembolic (TE) events

Reviewer #2: The manuscript of Calvet et al. evaluates the coagulation profile according to both traditional coagulation tests and thromboelastometry parameters in a group of 133 patients admitted for acute COVID-19 pneumonia. Interestingly enough the authors tried to find a correlation between the alterations of coagulation parameters and a composite endpoint "intubation and / or death in intensive care unit". The study results demonstrate a possible association between hyperfibrinogenemia or fibrinolysis shutdown on day 4 and the composite outcome "intubation and / or death in intensive care unit".

The results presented in the paper support the study's conclusions. Statistical analysis is appropriate and rigorous.

My comments:

Abstract: "despite high use of therapeutic anticoagulation" I do not agree with this concept since both traditional coagulative parameters and ROTEM profiles are poorly influenced by low molecular weight heparin.

We understand your point and acknowledge that ROTEM tests are poorly influenced by LMWH. Consequently, we have removed this part of the sentence from the Abstract.

Introduction: the authors should explain in more detail the relationship between COVID-19 related pneumonia, hypercoagulability and hypofibrinolysis

Thank you for this excellent remark. We now added this sentence at the beginning of the introduction.

“Patients with severe forms of COVID 19 have multifactorial hypercoagulability: increased D-dimer, fibrinogen, factor VIII levels, decreased protein C, protein S and antithrombin levels, platelet hyperaggregability, endothelial damage by Sars Cov 2 [1–3], and hypofibrinolysis. This hypercoagulability leads to thromboembolic events (TE) concerning macro and micro circulation [4–7]. These TE events, notably microcirculation thrombosis and pulmonary embolism, are one of the leading causes of multiple organ failure in critically ill COVID-19 patients [4,6,8–14].”

Methods: the authors should better explain the principle of thromboelastometry and of the assays used.

Thank you for the comment. The request has been taken into account (see reply above to the Academic Editor) 

Discussion: line 3 pag. 13 "Finally, we observed prolonged clot initiation on day 1 presumably owing to heparin exposure" see my comment above in the "abstract section".

Thank you for this excellent remark. The Discussion has been changed accordingly. 

“In our study, we observed prolonged clot initiation. Whether it can be related to heparin exposure requires a careful analysis since the impact of heparin on INTEM-CT depends on both the type of heparin and its dosage. While ROTEM tests are able to detect the presence of UFH on the basis of an increase in the difference, or in the ratio, between INTEM-CT and HEPTEM-CT, and whereas in patients administered with UFH a prolonged INTEM-CT is reported for heparinemia > 0.1 U/l, both traditional coagulative parameters and ROTEM profiles are poorly influenced by LMWH, and an increased INTEM-CT in patients receiving LMWH is only observed for heparinemia >0.4UI/l[69,70]. We observed similar results in our study. INTEM/HEPTEM CT > 1 was detected in patients receiving UFH for HDPA or CDA, and in 90% of patients under LMWH for CDA, but only in half of the patients under LMWH for HDPA (S12 Table).”

Table S10: Anticoagulation and prolonged CT on day 4 

At day 4 INTEM/HEPTEM CT < 1 INTEM/HEPTEM CT > 1 Pval

No anticoagulation 2 (11.1) 2 (4.5) 0.02

Preventive LMWH 0 (0) 1 (2.3) .

LMWH HDPA 13 (72.2) 15 (34.1) .

LMWH CDA 2 (11.1) 13 (29.5) .

UFH HDPA 0 (0) 3 (6.8) .

UFH CDA 0 (0) 10 (22.7) .

Direct Oral Anticoagulants 1 (5.6) 0 (0) .

Anti-vitamin K 0 (0) 0 (0) .

HDPA: high dose preventive anticoagulation; CDA: curative dose anticoagulation

---

## [Decision Letter · Decision Letter 1]

31 Oct 2022

Hypercoagulability in critically ill patients with COVID 19, an observational prospective study.

PONE-D-22-24018R1

Dear Dr. Dupuis,

We’re pleased to inform you that your manuscript has been judged scientifically suitable for publication and will be formally accepted for publication once it meets all outstanding technical requirements.

Kind regards,

Samuele Ceruti

Academic Editor

PLOS ONE

Reviewer's Responses to Questions

**Comments to the Author**

1. If the authors have adequately addressed your comments raised in a previous round of review and you feel that this manuscript is now acceptable for publication, you may indicate that here to bypass the “Comments to the Author” section, enter your conflict of interest statement in the “Confidential to Editor” section, and submit your "Accept" recommendation.

Reviewer #1: All comments have been addressed

Reviewer #2: (No Response)

2. Is the manuscript technically sound, and do the data support the conclusions?

Reviewer #1: Yes

Reviewer #2: (No Response)

3. Has the statistical analysis been performed appropriately and rigorously? 

Reviewer #1: Yes

Reviewer #2: (No Response)

4. Have the authors made all data underlying the findings in their manuscript fully available?

Reviewer #1: Yes

Reviewer #2: (No Response)

5. Is the manuscript presented in an intelligible fashion and written in standard English?

Reviewer #1: Yes

Reviewer #2: (No Response)

6. Review Comments to the Author

Reviewer #1: The authors of the paper answered all my questions in a timely and exhaustive manner. The manuscript has been improved and now, according to my judgement, it is ready to be published as it is.

Reviewer #2: The authors answered completely to the points that I raised. I suggest to accept the manuscript in the present form.

7. PLOS authors have the option to publish the peer review history of their article (what does this mean?). If published, this will include your full peer review and any attached files.

Reviewer #1: No

Reviewer #2: No

---

## [Editor Report · Acceptance letter]

2 Nov 2022

PONE-D-22-24018R1 

Hypercoagulability in critically ill patients with COVID 19, an observational prospective study. 

Dear Dr. Dupuis:

I'm pleased to inform you that your manuscript has been deemed suitable for publication in PLOS ONE. Congratulations! Your manuscript is now with our production department. 

Kind regards, 

on behalf of

Dr. Samuele Ceruti 

Academic Editor

PLOS ONE